# Personalized Dosimetry in the Context of Radioiodine Therapy for Differentiated Thyroid Cancer

**DOI:** 10.3390/diagnostics12071763

**Published:** 2022-07-21

**Authors:** Massimiliano Pacilio, Miriam Conte, Viviana Frantellizzi, Maria Silvia De Feo, Antonio Rosario Pisani, Andrea Marongiu, Susanna Nuvoli, Giuseppe Rubini, Angela Spanu, Giuseppe De Vincentis

**Affiliations:** 1Medical Physics Department, “Policlinico Umberto I” University Hospital, 00161 Rome, Italy; m.pacilio@policlinicoumberto1.it; 2Department of Radiological Sciences, Oncology and Anatomo-Pathology, Sapienza University of Rome, 00161 Rome, Italy; miriam.conte@uniroma1.it (M.C.); mariasilvia.defeo@uniroma1.it (M.S.D.F.); giuseppe.devincentis@uniroma1.it (G.D.V.); 3Section of Nuclear Medicine, Interdisciplinary Department of Medicine, University of Bari Aldo Moro, 70121 Bari, Italy; apisani71@libero.it (A.R.P.); giuseppe.rubini@uniba.it (G.R.); 4Unit of Nuclear Medicine, Department of Medical, Surgical and Experimental Sciences, University of Sassari, 07100 Sassari, Italy; amarongiu@uniss.it (A.M.); snuvoli@uniss.it (S.N.); aspanu@uniss.it (A.S.)

**Keywords:** Differentiated Thyroid Cancer, personalized dosimetry, radioiodine therapy, absorbed dose

## Abstract

The most frequent thyroid cancer is Differentiated Thyroid Cancer (DTC) representing more than 95% of cases. A suitable choice for the treatment of DTC is the systemic administration of 131-sodium or potassium iodide. It is an effective tool used for the irradiation of thyroid remnants, microscopic DTC, other nonresectable or incompletely resectable DTC, or all the cited purposes. Dosimetry represents a valid tool that permits a tailored therapy to be obtained, sparing healthy tissue and so minimizing potential damages to at-risk organs. Absorbed dose represents a reliable indicator of biological response due to its correlation to tissue irradiation effects. The present paper aims to focus attention on iodine therapy for DTC treatment and has developed due to the urgent need for standardization in procedures, since no unique approaches are available. This review aims to summarize new proposals for a dosimetry-based therapy and so explore new alternatives that could provide the possibility to achieve more tailored therapies, minimizing the possible side effects of radioiodine therapy for Differentiated Thyroid Cancer.

## 1. Introduction

The most frequent thyroid cancer is Differentiated Thyroid Cancer (DTC) representing more than 95% of cases [1]. It had become the most common cancer in women by the year 2020 after breast cancer, colorectum cancer, lung, and cervix uteri cancer [GLOBOCAN2020]. Although it represents only 1% of all human malignancies, it has been found in 5% of thyroid nodules, and in turn, they have a 20–30% prevalence in the general population [2]. The histologic derivation is the follicular epithelium from which it maintains the histological characteristics such as sodium iodide symporter (NIS) expression [3]. The thyroid follicular epithelial cells represent the embryologic derivation of DTC. Classically, it can be divided into three histological types: papillary thyroid cancer (also known as PTC, the most common), follicular thyroid cancer (FTC, the second most frequent), and Hürthle cell thyroid cancer. The last one was considered a particular type of FTC, but for the significant histopathological and molecular differences it was classified as an independent tumor type in 2017 by World Health Organization. Hürthle cell carcinoma is so defined as a derivation of follicular thyroid cell carcinoma and not a variant [4].

Even if the causes of thyroid cancer are unknown, many studies have evaluated risk factors. High risk factors in developing DTC are head and neck region radiation exposure, chromosomal alterations such RAS and BRAF genes mutations and PAX8/PPARγ fusion protein expression, and hereditary conditions such as medullary thyroid cancer, syndromic and non-syndromic familial non-medullary thyroid cancer. Low risk factors are thyroid imaging with iodine 131 (^131^I), iodine deficiency, high thyroid stimulating hormone (TSH) level, autoimmunity, the presence of thyroid nodules, and the environmental pollutants, but also lifestyle, diet, and high BMI. It is not clear if estrogens have a role in increasing risk: some studies reported an increasing risk associated with exogenous estrogen, while loss of ovarian estrogen lowers it. Estradiol is seen as a stimulator for benign and malignant neoplasm [5,6]. Lastly, but not for importance, the histological type, in particular tall cell cancer and columnar cell cancer, but also the vascular, lymphatic invasion, and distant metastasis are considered risk factors that contribute to poor prognosis [7].

All variants of DTC can be associated with genetic alterations: in PTC the BRAFV600E point mutation is the most frequent genetic alteration involving 45–60% of patients, followed by RAS and TERT promoter point mutations. In FTC, the most common alteration is RAS mutation (25%), while in Hürthle cell cancer, frequent mutations are in complex I mitochondrial DNA. In poorly differentiated thyroid cancers, there are a huge variety of genetic alterations, especially in BRAF and RAS genes [8,9,10]. The prompt treatment gives a generally excellent prognosis for patients affected by DTC and even if the 10-year survival rate in patients with distant metastasis is approximately 25–40% [11,12,13], the 10-year overall cause-specific survival is 85% [14,15]. A suitable choice for the treatment of DTC is the systemic administration of 131- sodium or potassium iodide, also known as radioiodine therapy, RIT, or RAIT. It is an effective tool used for the irradiation of thyroid remnants, microscopic DTC, other nonresectable or incompletely resectable DTC, or all the cited purposes. The present paper aims to focus attention on iodine therapy for DTC treatment and has developed due to the urgent need for standardization in procedures. Nowadays, there is more and more necessity for studies that could determine which procedure is the best choice since no unique approaches are available. This review aims to summarize new proposals for a dosimetry-based therapy and so explore new alternatives that could provide the possibility to achieve more tailored therapies, minimizing the possible side effects of RAIT.

## 2. Risk Classification

As regards the prognosis, the risk of cancer-related death for DTC has been estimated by TMN classification. It was established in V classes: group I includes patients younger than 55 years old with any T and N, Mo and cancer-related death risk is <2%. Class II represents patients with any T and N but M1 or patients with an age major or equal to 55 years and TMN classification T1/T2 N1 M0 or T3a/T3b, any N and M0. In this group, the risk is equal to 5%. Group III included patients at least 55 years old with T4a, any N, and M0 with a risk of 5–20%. IVa class identifies subjects at least 55 years with T4b, any N, M0, and a risk > 50%. IVb class has a risk of death > 80%, with any T, any N, and M1 [16,17] (See Table 1).

Hürthle cell, invasive FTC, tall cell, diffuse sclerosis variant, and columnar cell carcinomas are characterized by the worst prognosis due to their high vascularity and therefore a higher risk of hematogenic spread. It is common in patients with these histological types to have metastases at diagnosis [18]. Risk classification is a milestone for iodine treatment. The therapy of DTC is customized on the basis of histological classification and Tumor Nodes Metastasis (TNM) staging and risk of recurrence. In particular, the risk of structural recurrence has been valued by the American Thyroid Association (ATA) on the basis of histological characteristics such as capsular, lymph nodes, and vascular invasion, the presence of aggressive histologic type, the number of lymph nodes metastasis, and thyroglobulin serum levels. The patients are, therefore, classified at a low (recurrence risk <5%, it is the case of PTC and well-differentiated and minimally invasive FTC), intermediate (5–20%), or high (>20%) risk of recurrence [15] (See Table 2).

More than 80% of patients are identified as low risk [19]. The risk influences therapy management: for ATA iodine therapy is not recommended for the residual ablation in low-risk DTC, while adjuvant therapy is recommended in intermediate-risk with activities between 30 mCi (1110 MBq) and 150 mCi (5550 MBq). In 70-year-old or older patients, activities up to 200 mCi (7400 MBq) of ^131^I are desirable to avoid possible adverse events.

## 3. Indications for ^131^I-Therapy

### 3.1. EAMN Recommendations

European Association of Nuclear Medicine (EANM) guidelines recommended ^131^I therapy only for the ablation of thyroid remnants as a post-surgical adjuvant procedure, for microscopic or incompletely resectable DTC, other non-resectable lesions, or both purposes (See Table 3).

The administration of RAIT also for ATA and EANM ought to be limited to remnant ablation, adjuvant treatment, or treatment of known diseases such as locoregional metastasis and distant metastasis [20]. In particular, RAIT is recommended in microscopic DTC. It finds explication in better results of RAIT in microscopic or small macroscopic tumors than in larger lesions [12]. The intention of cure or palliation using RAIT should be considered case by case, which represents the best choice as adjuvant treatment post-surgery of persistent or recurrent DTC until the lesion is iodine avid. Lymph nodes, lung, and most tissue metastasis can be successfully treated by RAIT with or without the auxilium of surgery, while bone and brain metastasis does not have a high rate of cure [21]. Moreover, less differentiated RAIT tumor histotypes, such as papillary tall-cell, columnar cell or diffuse sclerosing or follicular widely invasive, poorly differentiated, or Hürthle cells, have a greater risk of relapse and a lower overall survival but, despite diminished NIS expression, such tumors may be responders to RAIT [22]. RAIT should be also considered in patients >45 years since the most aggressive forms are diffused in older age and have a reduced overall survival [15], but also in patients unable to tolerate surgery or other eventual other therapies such as chemotherapy. According to recent European guidelines, in addition to pregnancy and breastfeeding, exclusions for RAIT are patients with unifocal papillary thyroid cancer ≤1 cm sized without metastasis, thyroid capsule invasion, or a history of radiation exposure. Additionally, tall-cell, columnar cell, or diffuse sclerosing subtypes are excluded [3]. Relative exceptions are bone marrow depression, pulmonary function restriction, especially if lung metastasis is present, and salivary gland function restriction. It is also necessary to evaluate the presence of neurological symptoms or damage since inflammation and local edema due to RAIT effect on cerebral metastasis could result in severe compression effects. Furthermore, 18F-fluorodeoxyglucose (18F-FDG) uptake in metastases is an evident indicator of the presence of radioiodine non-avid disease and a considerable independent unfavorable prognostic indicator. Specifically, the numbers of FDG-avid lesions and the higher maximum standard uptake values (SUVmax) correlate with overall mortality [23]. Radioiodine therapy is commonly well tolerated even if it has been suggested to be a potential second primary malignancy superior to 30% in a study conducted on 13 types of cancers [24]. However, EAMN evaluated that risk to be inferior to 1% in a latency period of ≥5 years. More cases have been observed when cumulative radioiodine activities exceed 20–30 GBq [25,26]. To maximize therapeutic effects and minimize collateral events, the maximum activity allowed has been proposed. A single administration of 1–5 GBq is recommended for the ablation of post-surgical thyroid remnants, even if there are different clinical practices among centers. It remains controversial which activities between 1.11GBq, 1.85 GBq, or 3.7 GBq are the best option. A systematic review by Hackshaw et al. established that ablation success rates are similar using 1.11 GBq and 3.78 GBq of ^131^I [27]. The accepted dosimetry approach is the blood-based method that takes into account blood and the whole body as compartments under the assumption that a similar iodine concentration has been found in blood and red marrow [28]. Still, the presence of extended metastatic bone involvement represents a limitation for the method due to the underestimation of the absorbed dose in red marrow. Likewise, in the presence of diffuse pulmonary micro-metastases, this method should be applied with caution because the critical organ, in this case, is the lung and not red marrow so dosimetry could be applied to the lung [29,30].

### 3.2. ATA Recommendations

For ATA, adjuvant treatment after total thyroidectomy should be deemed in intermediate-risk and high-risk DTC using activities between 30 mCi (1110 MBq) and 150 mCi (5550 MBq) [31]. However, a common practice was determining the activity through empiric fixed doses chosen by physicians on the basis of experiences, convention, and patient parameters [32,33]. The method mostly used was Beierwaltes’s proposal [34]. It considers that the administered activity should not exceed 5.55–6.2 GBq for regional nodes that could not be removed by surgery, 6.2–7.4 GBq for pulmonary metastasis, and 7.4 GBq for skeletal metastasis. Other intermediate activities were used such as Schlumberger et al. [35], which used an initial activity of 3.7 GBq of ^131^I for pulmonary and bone metastasis every 3–6 months, and Menzel et al., which used a fixed activity of 11.1 GBq at 3-month intervals [36]. In this regard, a riveting study by Iizuka and collaborators makes a comparison between clinical outcomes of intermediate-high risk patients treated with high-dose and low-dose iodine therapy [37]. For low-dose therapy is intended the administration of 1100 MBq of ^131^I, while the high-dose is defined as 2960–3700 MBq of ^131^I administration. They collocated initial success with a thyroglobulin (Tg) level inferior to 2.0 ng/mL without thyroid-stimulating hormone administration and no ^131^I uptake in the thyroid region at ^131^I scintigraphy performed 6–12 months after RAIT. The conducted analysis established that the success rate of iodine therapy tended to be superior in high-risk patients treated with high-dose therapy even if no statistical differences have been found between the two groups in terms of success rate. These results reveal that in high-risk patients, a high-dose treatment is preferable. On the other side, Qu et al. evaluated the therapeutic response between low- and intermediate-risk patients treated with high-activity (3.7 MBq) and low-activity (1.1 MBq) [38]. The ablation and therapeutic results were similar between the two groups. Lymph node involvement and serum Tg seemed to influence ablation and therapeutic response, respectively. The success rate of ablation was lower for patients in the N1b stage than for patients in the N0 stage, while increased with lower serum Tg levels. In particular, they noted that a pre-treatment Tg level was associated with a higher better response: a level of 0.47 μg/L was a cut-off for predicting ablation results and a level of 3.09 μg/L for predicting therapeutic response. Thirty-two years of follow-up in DTC patients after RAIT was conducted by Martins-Filho and colleagues, to evaluate tumor behavior in correlation with cumulative iodine doses and survival. Patients with higher Tg serum levels had progressive disease and needed more frequent and higher doses of iodine treatment [39]. They observed that a patient with an age below 45 years had a 70% chance of complete remission with a cumulative activity of 100 mCi. If the activity ranges in 1 Ci, there is a 27% chance of stabilizing the disease. A high chance of progression was evident for cumulative activities of 600 mCi in a patient with an age above 45 years, and also for cumulative activities of 800 mCi or higher in a patient under 45 years. The data suggest a careful evaluation of further RAIT. There also is no common line of decision making for children since some centers set the activity by body weight or surface area or by age [40], while for German guidelines it should be adjusted according to the 24 h thyroid bed uptake and body weight: if uptake is <5%, the activity should be 50 MBq/kg, 5–10% uptake would assure an activity of 25 MBq/kg and 10–20% uptake an activity of 15 MBq/kg [41].

### 3.3. RAIT in Metastatic DTC

A field that needs standardized applications again is RAIT in metastatic DTC. There had been multiple differentiated approaches in the past and present among centers. First of all, the empiric proposals: Beierwalters et al. provided 6.48 GBq (175 mCi) for lung and 7.4 GBq (200 mCi) for bone [34]; Schlunmberger et al. used 3.7 GBq (100 mCi) every 3 months to 6 months until whole-body scan (WBS) was negative [31]; Petrich et al. used a therapeutic activity of 3.7 GBq (100 mCi) and if metastasis were still evident, they retreated the patient with 7.4 GBq (200 mCi) [42]; Brown et al. used 5.5 GBq (150mCi) every 3–4 months until the scan was negative of there was progression; Menzel et al. administered 11.1 GBq (300 mCi) every 3 months [36]; Durante et al. opted for 3.7 GBq (100 mCi) every 3–9 months during the first 2 years, once a year after until no metastasis was evident [12]; Hindle et al. used 3.7 GBq (100 mCi) every 6 months if there was lung uptake and chest radiography was negative, but if a cumulative activity of 18.5 GBq (500 mCi) was reached and lung uptake was present, reduced the therapy once a year and then every 2 years [43]. Again, Hindle et al. recommended the use of activity between 3.7 and 5.5 GBq (100–150 mCi) every 6 months when lung uptake was evident but also chest radiography was [44]. Ghachem et al. compared the ablation rate between patients treated with 1.1 to 1.85 GBq and patients treated with 3.7 GBq of ^131^I [45]. They had a similar ablation rate but the likelihood to have remission was 1.83 times greater for higher activity determining that mini dose protocol is not more effective in ablation than the higher dose protocol. 2013/59 EURATOM recommended for metastatic DTC that the optimal activity is the one that permits the best response rate and low toxicity even if it is still in debate and the same international guidelines do not suggest if it is preferable to use empiric or dosimetry-based therapies in these patients [46].

## 4. Collateral Effects of ^131^I Therapy

### 4.1. Bone Marrow Deficiency

The side effects of RAIT, especially on bone marrow, have been widely described even if the great majority of these studies do not evaluate them through dosimetry studies. In an intriguing review by Andresen et al. [47], adverse effects have been related to dose levels summarizing what was reported in the literature.

To exclude the possibility to have bone marrow damage, the absorbed dose to the blood should not exceed 2 Gy, and the whole-body retention after 48 h from the administration should not be superior to 4.4 or 3 GBq in the absence or presence of iodine-avid diffuse lung metastases, respectively [48,49].

### 4.2. Nausea, Neck Pain, Lacrimal and Salivary Disfuntion

The considered side effects such as nausea, neck pain, lacrimal and salivary dysfunction, and altered smell and taste seem to be acute effects that are more frequent in therapies with 100 mCi, and less frequent for dosages between 30 and 50 mCi (respectively, 13% vs. 4%, 17% vs. 7%, 10–24% vs. 8–20%, 5–16% vs. 6–13%, 6% vs. 0 and 2% vs. 0). Sialadenitis has a frequency between 2 and 67% and nasolacrimal duct obstruction 3.4% [50].

RAIT should be carefully evaluated in patients with restriction of salivary to minimize side effects (e.g., xerostomia) since salivary glands are the physiological site of iodine uptake. It happens because of the presence of NIS, a plasma membrane glycoprotein, that mediates active I(-) transport in various organs such as the thyroid, the previously written salivary glands, stomach, lactating breast, and small intestine [51]. Acute salivary gland swelling and pain decrease a few days after ^131^I-therapy in patients with DTC but, in some patients, the onset of these symptoms is slower. In other patients, the symptoms could lead to chronic radiation sialadenitis. These complications are clinically significant in 10–30% of subjects even if their frequency is not well known [52,53,54,55,56,57,58]. In particular, in the prospective cohort study by Hyer et al., patients were treated with an ablative activity of 3GBq and a further of 5.5 GBq in case of residual disease. Of the 79 patients examined, 26% developed salivary gland toxicity after iodine therapy, with a median onset of 48 h after ^131^I administration and a median duration of symptoms of 12 months. The reported symptoms were xerostomia in all cases, 2 patients were referred with an episode of swelling and pain, 6 patients had two episodes, and 8 subjects had three or more episodes. The submandibular gland was interested in 50% of subjects, parotid in 39%, and 11% of patients had both glands affected. Among these patients, 55% had lymph node metastasis while only 32% of patients without salivary gland involvement had lymphatic metastasis. The mean cumulative activities were 8.57 GBq in patients with salivary complications, 9.04 GBq for the others. These results demonstrated that repeated activities were more associated with the development of salivary gland toxicity. Conversely, cumulative doses seemed not to be related suggesting how it is more damaging repeated doses than a single dose [59]. Since sialadenitis is not always clinically evident, many other studies using 99mTC-pertechnetate scintigraphy were conducted to evaluate salivary gland function. Abnormalities at salivary gland scintigraphy were noted with activities of more than 18.5 GBq ^131^I [60], but also with activities between 3.7 GBq and 38.7 GBq of ^131^I [61]. As regards the timing of symptoms that arise, it was reported from months to years after RAIT [58,62], and in a study by Allweiss et al. [57], they continued up to 2.5 years later. It was also reported an increase of risk in developing salivary gland neoplasm such as pleomorphic adenoma, non-Hodgkin’s lymphoma, and mucoepidermoid carcinoma and it is proportional to the ^131^I dose [60]. Moreover, it seems that the administration of lemon candy post-therapy increases the risk of acute and chronic salivary gland toxicity. The rationale for using lemon candy after iodine administration is based on the concept that ascorbic acid increases the salivary flow and then iodine clearance. On the other flip, it enhances blood flow to rising ^131^I uptake. In the context of diminished glomerular filtration rate determined by hypothyroidism, the serum level of ^131^I is high. Additionally, the remnant or metastatic tissues are not so avid as the normal thyroid. Consequently, the continuous consumption of lemon candy could only improve this condition leading to a greater amount of dose to the salivary gland [63].

### 4.3. Sexual Sphere Side Effects

Male and female infertility have subacute latency, differently from the upper cited sialadenitis but also from nasolacrimal duct obstruction, and the onset of secondary malignancy that are late effects that seem to be evident only for ^131^I activity major to 100 mCi. Notably, in males, a transient decrease of follicle-stimulating hormone (FSH) levels and reduced sperm motility were evident while in women it has been observed lower birth rate in the age group between 35 and 39 years.

A potential impact has been pointed out by Bourcigaux et al. on exocrine and endocrine testicular function after 3.7 GBq iodine administration. They observed Sertoli cell function and induced sperm chromosomal abnormalities 3 months and 13 months after therapy [64].

### 4.4. Second Recurrences

According to literature from 30 to 40 years old, recurrences were thought to occur 10–20 years after RAIT. Today, thyroid cancer is presented at an earlier stage and has a lower recurrence rate of 3–10% [51,52,53,54]. Moreover, Carrillo et al. demonstrated that patients with DTC treated with ^131^I, which have a biochemical recurrence and no diagnostic WBS (DWBS) or extensive studies, have lower frequencies of second recurrence (SR) compared to the group that, prior RAIT, underwent a DWBS with 5 mCi of ^131^I and Magnetic Resonance Imaging (MRI) and/or 18F-FDG PET if DWBS was negative [55]. In a study by Rubino et al., second primary malignancies (SPM) and leukemia risk were evaluated. They selected a cohort of 6841 Swedish, Italian, and French patients with FTC or PTC diagnosed during the period 1934 and 1995 [25]. In total, 576 patients had SPM and among them, 13 presented a third malignant neoplasm less than 2 years after. The risk of cancer was globally increased by 27% compared to the general population of the same countries. The mean time between thyroid cancer diagnosis and SPM presentation was 15 years. No gender differences were observed. The cancer types with a significantly increased risk were cancers in the digestive tract, bone and soft tissue, skin melanoma, kidney, central nervous system, endocrine glands (excluded thyroid), leukemias, female breast cancer, and genital male malignancies. No association between SPM and external radiotherapy was observed, so no interactions between them could be described. In total, 13 cases of oral cancer were counted, and of them, 7 were caused by ^131^I administration. They also demonstrated a significant association between cumulative activities of ^131^I and the risk of solid cancer, with an increase of 4% per GBq.

### 4.5. The Stunning Fenomenon

Another noteworthy event to take into consideration for RAIT success is represented by the “stunning” phenomenon. It consists of the diminishing of iodine uptake due to the suboptimal therapeutic effects, biological effects, or both, caused by a previous diagnostic radioiodine administration. To avoid this phenomenon, it is suggested to shun unnecessary pre-therapeutic ^131^I administration, when the indications to RAIT are clear, or if necessary, to use lower radioiodine activities before RAIT. Hence, an activity of 3–10 MBq is recommended for uptake quantification and 10–185 MBq for WBS. An alternative could be opting for other tracers such as 123iodine (^123^I), 99mTechnetium (99mTc), or ^124^I in PET/CT. Dosimetry studies demonstrated that activities of 10–20 MBq represent a considerable radiation burden for thyroid cells. The stunning effect seems to be caused by direct radiation damage to thyrocytes [65,66,67,68,69,70,71,72,73]. This results in a diminishing in iodine transportation by 50% also with an absorbed dose of 3 Gy [74] and down-regulation in NIS expression [75]. These phenomena lead to a diminishing in iodine uptake [69]. In this regard, Verburg and collaborators compared two groups of patients, one undergone a pre-ablative 24 h uptake measurement with 40 MBq ^131^I and the other without pre-ablative diagnostic scintigraphy [65]. They demonstrated how the success rate of ablation was 2 times greater in the group without pre-ablative scintigraphy than in the group that underwent a pre-therapeutic uptake study. Similar results were cited in the papers: Jeevanram et al., who reported a decrease of 20–25% uptake when the activity of 111–185 MBq was used for pre-ablation scan [76]; Lassman et al., after 74 MBq [66]; Muratet et al., for 111 MBq [77]; Hu et al. and Lees et al., after using 185 MBq [67,78]; and Park et al., for 370 MBq [79]. No stunning events were observed in groups undergoing 37 MBq scans [77], with ^123^I or without any pre-therapeutic scans [78].

### 4.6. Reduced Kidney Function and Dosimetry Considerations

Notably, patients with reduced kidney function or hemodialysis reached higher bone marrow doses per unit of administered activity [33]. Recently, in low-risk cancer, the long-term cancer control rates were equivalent if therapeutic activities between 30–50 mCi or major or equal to 100 mCi were used [80]. In contrast, it was suggested that higher doses could better control cancer in intermediate- and high-risk patients [47,81]. In a recent phase III trials, Schlumberger et al. proved equivalent rates of thyroid remnant ablation with 30 versus 100 mCi [82]. Mallick et al., instead, asserted that low-dose using recombinant human thyrotropin alfa was as effective as high-dose radioiodine, but it has a lower rate of side effects [83]. In a paper by Khang et al., the mortality risk for SR is higher for patients with a cumulative ^131^I activity > 37 GBq. On the other hand, the risk of DTC-specific mortality is lower compared to the group which received a cumulative activity < 37 GBq and has similar all-cause mortality to low-activity RAIT patients [84]. In a study by Klubo-Gwiezdzinska et al., it was well demonstrated that the dosimetry approach has more efficacy compared to the empiric approach [85]. These considerations highlight how the dosimetry-based approach is more effective and represent a key role for more personalized therapy, especially in patients with metastatic disease to minimize side effects.

## 5. Personalized Dosimetry in Cancer Radiation and Nuclear Medicine Therapy (State-of-the-Art and General Regulatory Requirements)

### 5.1. Dosimetry

The consolidated experience and wide availability of follow-up data on treated patients in radiotherapy demonstrated that the absorbed dose (defined as the energy absorbed per unit mass of tissue) is well-correlated to the effects of tissue irradiation, so it is a reliable indicator to predict biological response. Personalized dosimetry has been a standard procedure for many years in most radiation therapies, such as external beam radiation therapy, brachytherapy, and proton- and hadron-therapy. “Optimization” of treatment means individual planning of the treatment, also known (and hereinafter named) “treatment planning”. Treatment planning is based on a patient model defined by the patient medical images, and on the related individualized dosimetric calculations. For example, radiotherapy treatment planning is performed by optimizing the physical/geometrical parameters of the irradiation obtained from a linear accelerator (dose rate, gantry angle and/or movement, the geometrical configuration of a multi-leaf collimator, no. of monitor units, etc.), aiming at the best balance, between irradiation of target volumes (efficacy) and sparing of healthy tissues (toxicity). Numerous dedicated software (called treatment planning systems) has been commercially available for many years to aid personalized dosimetry. Additionally, for Nuclear Medicine Therapy (NMT), the mechanism of action involved is radiation-induced cell death, so the absorbed dose accounts for the biological effects of the treatment, and this also occurs with radiopharmaceuticals other than iodine [86,87,88,89]. Individualized dosimetry is needed due to inter-patient variability in the biokinetics of organs and lesions. Indeed, large variations in absorbed dose were observed also for patients having similar clinical conditions and administered with the same activity, leading to different clinical outcomes [90,91]. Optimization in NMT implies the dosimetry-based calculation of the activity to be administered to guarantee adequate irradiation of the target, safeguarding the organs at risk as much as possible, so patient-specific calculations of absorbed dose to lesions and organs are equally required. The use of dosimetry in nuclear medicine started out of necessity to assess the potential risks of diagnostic imaging agents, so its early implementation was based on an anatomical geometry designed to represent the average patient of a population, rather than a specific individual [92]. The Medical Internal Radiation Dose (MIRD) Committee re-formulated the internal dosimetry basic equations as a product of two quantities, one that depends on the pharmacokinetics of the radionuclide in the body and one that is a property of the radionuclide and the representative patient anatomy used for the calculation [93,94]. The first term is the total number of nuclear transformations (or several decays) of the radionuclide in a given source tissue/organ, experimentally derived by individual biokinetics sampling; the second term consists of a dosimetric factor accounting for the absorbed dose per decay for a given source-target pair, pre-calculated on an “average” individual. This approach allowed for tailored biokinetics estimation providing the basis for personalized dosimetry in NMT; however, it was dedicated to radiation risk prevention in diagnostic imaging (morphology was not individualized, and tumor description was not included) so not appropriate to evaluate possible toxicity and efficacy of NMT treatments. Further improvements consisted of including a patient-specific complete description of the individual to be treated. Increased availability of SPECT-CT and PET-CT scanners and consequently improved image quality allowed for a more direct approach, using the measured patient activity distribution from PET/CT or SPECT/CT images, superimposed over the anatomy as obtained by the CT portion of the imaging scan. Such voxelized dosimetry approaches use Monte Carlo or point-kernel methods to calculate maps of the spatial distribution of absorbed dose [95,96,97,98,99,100]. Detailed descriptions of how these measurements should be made using quantitative imaging methods have been extensively treated and reported in the literature [101,102]. Recent technological improvements such as the increased availability of hybrid imaging systems and an increase in computational efficiency may lead to a rational, absorbed dose-driven approach in NMT, in principle allowing for combination with radiotherapy [103,104]. Personalized dosimetry in NMT may be described as equivalent to a real-time pharmacodynamic study, performed pre-treatment on patients who have been administered with a tracer dose of the therapeutic radiopharmaceutical (or another equivalent to it, useful for imaging). Dosimetric calculations are more demanding with respect to other radiation therapies, as a pre-treatment biokinetics study after administration of a tracer activity requires: (i) preliminary calibrations of instrumentations (to perform in vivo repeated measurements for biokinetics assessments), (ii) availability of technologist and scanner time for acquiring repeated scans, (iii) good patient compliance, (iv) patient modeling and dedicated software for dosimetric calculations and (v) related know-how. Further technical aspects of biokinetic studies and dosimetric calculations are defined according to the type of NMT treatment considered; however, frequent patient involvement in experimental acquisitions and availability of technologist and scanner time for acquiring repeated scans are generally distinctive features of treatment planning in NMT. Differently from other radiation therapy modalities, treatment planning software has been made available on the market in recent years only for some types of treatment (e.g., hepatic radioembolization with labeled microspheres and peptide receptors radionuclide therapy). In addition to the general logistical, technical, and clinical aspects, a further specific technical problem may arise from the limitation in spatial resolution of the instrumentation used in current nuclear medicine imaging, sometimes preventing an accurate dosimetric evaluation: e.g., the difficulty to measure accurately the mass of small target and consequently to calculate the absorbed dose, makes the dosimetry implementation difficult, to the point of becoming impossible. The main examples are patients with high tumoral marker levels with negative nuclear medicine imaging (micro-metastases from DTC or neuroblastoma).

### 5.2. Dosimetry in Radioiodine Therapy for DTC

In radioiodine therapy for DTC, limitations of spatial resolution in SPECT drastically affect biokinetics and mass calculations on remnants after thyroidectomy, on neck lymph nodes, or small metastases. Additional, possible issues when using therapeutic radionuclides of not having useful photon emission for imaging could derive from the unavailability of an adequate radiopharmaceutical for predictive dosimetry. The knowledge of the absorbed dose thresholds associated with various clinical effects is also essential for treatment planning, as it essentially consists of an appropriate cost–benefit analysis of the treatment based on the knowledge of dose-thresholds associated with given probabilities of side effects and tumor control. Several follow up studies have been conducted in recent decades making a large amount of information available for radiation therapies using external sources [105], for tumors and organs at risk (OAR, as commonly called in radiotherapy healthy tissues/organs placed near the treatment target, whose irradiation could cause damage that would make changes to the radiotherapy treatment plan), but this information is often not applicable to the context of nuclear medicine therapy. The biological effects of a given absorbed dose to tumors and normal organs depend on the dose rate at which the irradiation occurred. In NMT, a given absorbed dose to tissues is delivered over a period of many weeks with an exponentially decreasing dose rate (due to biologic clearance of the radiopharmaceutical and physical decay of the radionuclide), this causes a very different effect from that of the same absorbed dose delivered at the much higher dose rates used in radiotherapy (around some Gy per minute), typically with fractionated treatments at 2 Gy fractions/day over a period of 15–30 days. The difference in biological outcome will depend on the biological repair and radiosensitivity properties of the irradiated tumor and tissues [106,107,108]. Useful information could be obtained also from follow-up studies associated with other radiation therapies, provided that adequate radiobiological models are available to perform the appropriate conversion of the absorbed dose into biologically effective dose for isoeffective dose calculations [109,110]. More frequently, dedicated studies are required to derive absorbed dose thresholds for NMT [111].

### 5.3. Regulatory

The European regulations (EU Directive 59/2013) consider now mandatory the treatment planning and verification in all patients’ exposures for radiotherapeutic purposes including also nuclear medicine therapy [112]. The new definition of “radiotherapeutic” contained in the directive clearly states, “pertaining to radiotherapy, including nuclear medicine for therapeutic purposes”. Moreover, Article 56 of the directive (“Optimization”) clearly states that “For all medical exposures of patients for radiotherapeutic purposes, exposures of target volumes shall be individually planned, and their delivery appropriately verified taking into account that doses to non-target volumes and tissues shall be as low as reasonably achievable and consistent with the intended radiotherapeutic purpose of the exposure”. This new directive clearly states that also NMT is a radiotherapy practice and must be subject to the same legislation and same principles and indications of good radiotherapy practice. In recent years, the member states have transposed the new directive into national legislation, and national regulatory institutes and scientific associations should define guidelines for the correct application of the regulatory provisions, as part of a correct cost–benefit analysis that considers the current scientific knowledge and state-of-the-art in clinical dosimetry, the technological equipment commonly available, the logistical sustainability of the activities [113]. Despite these important regulatory clarifications, which are also in line with current scientific knowledge, two further problems for the adoption of the treatment planning approach in NMT have arisen. The first one is the apparent legislative conflict between posology and individualized dosimetry [114], but scientific evidence on dose–response correlation (if or when available) should prevail, and the activity to be administered in NMT should be calculated according to the optimization principle of radiotherapy, without neglecting the pharmaceutical legislation for all the other pertinent factors [113]. The second problem arose parallelly in recent years from the new perspective of evidence-based medicine (EBM), which has brought about a paradigm shift not only in medical practice and education but also in study design and in the appraisal and classification of published research in medicine [115]. The principles created by pioneers in the field of EBM are now widely accepted as the standard not only for appraising the quality of evidence but also for evaluating the strength of evidence produced by research. Unfortunately, randomized dosimetric trials in NMT have scarcely been conducted, with just a recent exception for liver radioembolization [116]. Hence, data derived in the new EBM perspective, clearly affirming the superiority of a dosimetric approach and defining appropriate dose thresholds, would be considered still missing for most NMT methodologies. However, wide evidence from monocentric studies is reported in the literature on dose–response correlations, clearly affirming the superiority of the dosimetric approach, for instance for radioiodine therapy of DTC [33,87,117,118,119,120,121,122], for peptide receptor radionuclide therapy of neuroendocrine tumors [110,123,124,125,126], and liver radioembolization [127,128,129,130]. It is also worth noting the intrinsic limitations of comparative retrospective studies. After a recent retrospective multi-center comparison reported no overall survival (OS) difference of empiric versus dosimetry-based radioiodine therapy [131], an important discussion has arisen about the difficulty of performing multi-center retrospective analyses due to differences in treatment regimens and patient cohort characteristics among centers [132,133]. Any comparative study in DTC is difficult due to, among other things, the long observational periods required to get reliable data on long-term outcomes, and the relatively low number of advanced DTC because fortunately only a small fraction of patients progress to advanced disease. What has been said so far pushes to design and execute randomized trials, which unfortunately have not yet been carried out. However, it is appropriate to examine the main literature results available on personalized dosimetry, as this certainly helps in the design of new randomized trials. Moreover, it should also be specified that based on the data reported in the scientific literature (international peer-reviewed journals) or based on recommendations from national and international scientific societies, a therapeutic team could, in science and consciousness, overcome the indications of posology, performing treatment planning.

## 6. Personalized Dosimetry in Radioiodine Therapy for Differentiated Thyroid Carcinoma

Radioiodine therapy is a prototypical example of NMT and has been for many years a well-established practice for the treatment of hyperthyroidism and DTC to ablate remnant thyroid tissue and to treat iodine-avid metastases. The discoveries that iodine is concentrated via a NIS in the thyroid follicular cells and differentiated follicular neoplastic cells encouraged the early use and widespread of radioiodine for the treatment of patients with thyroid cancers [134,135,136]. Nowadays, RAIT remains recommended for the treatment of patients affected by metastatic DTC [3,137]. A general overview from a clinical viewpoint of the treatment procedures, benefits, and risks associated with radioiodine therapy of DTC was reported in the previous sections.

### 6.1. Bone Marrow and Lung Issues

The thresholds for the lesion-absorbed dose to achieve a high therapy response in RAIT have long been known and generally accepted: 300 Gy for thyroid remnants and 85 Gy for metastases [117,118]. The main Organ At Risk (OAR) is hematopoietic red marrow so the possible side effect (predictable with dosimetry) is hematological toxicity, but also, lung fibrosis may occur in patients affected by lung metastases, and it must be prevented. Hematological toxicity is correlated to a threshold of absorbed dose to the blood of 2 Gy [80,138], which has also been confirmed in a more recent study [139]: no high-grade bone marrow (BM) toxicity (i.e., grade 3, G3) was observed in patients, as predicted by administration of a ^124^I tracer to receive a blood dose of less than 2 Gy (120 patients), whereas in patients predicted to receive a blood dose of 2 Gy or more (range 2–3.43 Gy), G4 leukocytopenia (one patient), G3 granulocytopenia (one patient), G3 thrombocytopenia (two patients), and G3 lymphocytopenia (four patients) were reported. Similar outcomes were also confirmed by Bianchi et al. [140], who reported severe toxicity after administration of ≥1.7 Gy to the red marrow in six of seven patients, but with spontaneous recovery. Further recommendations, in addition to limiting the mean dose to 2 Gy, were as follows: at 48 h after administration, the whole-body retention does not exceed 4.4 or 3 GBq in the absence or presence of iodine avid diffuse lung metastases, respectively [80]. As regards lung fibrosis, it is advisable to limit the mean absorbed dose below 30 Gy, which corresponds to the absorbed dose of the external radiation therapy, which is associated with a 5% risk of severe damage within 5 y [122].

### 6.2. Personalized Dosimetry in Small Targets

Although dosimetric endpoints associated with possible clinical effects of the treatment had been known for a long time, there is often the difficulty in quantifying the mass of small targets, because targets in DTC are frequently small (thyroid remnant, lymph nodes, or metastatic lesions) with respect to the typical spatial resolution of SPECT imaging systems. Some retrospective studies were performed for verification of dose-thresholds, using simplifications or alternative strategies to overcome the limits of quantification for small targets [141,142], but the expected results have not always been obtained. An independent verification study of dose-thresholds for remnant ablation performed on a small group of patients (n = 23), treated with a fixed administration of 3 GBq, yielded a maximum voxel absorbed dose to thyroid remnants for successful ablation of 99 ± 128 Gy (range 12–570 Gy), whereas for patients with persistent uptake, was 25 ± 17 Gy (range 7–49 Gy), with a significant difference (*p* = 0.030) [143]. These absorbed dose values (calculated as maximum absorbed dose in the target) seem sensibly smaller than the theoretical dose-threshold of 300 Gy, probably due to experimental errors in the quantification of activity in the target by SPECT acquisitions: even though these dosimetric data seem independent from the target size (the maximum absorbed dose value in the target is not based on the assessment of target mass, differently from the mean absorbed dose) limited spatial resolution of emission tomography causes partial volume effects which drastically lower the count intensity in small objects, so decreasing the estimated cumulated activity and consequently the absorbed dose [143]. A lack of accuracy in assessing mass and biokinetics for small targets drastically affects target dosimetry and consequently the possibility to predict tumor control. However, a useful indirect dosimetric indicator for tumor control can also be determined.

### 6.3. Thyroid Remnant Ablation, Locally Advanced Disease and Distant Metastases

A retrospective study on thyroid remnant ablation including 449 DTC patients [144], assessed the success of ablation (defined as a negative diagnostic ^131^I whole-body scan and undetectable Tg levels at 6 months follow-up) as predicted by the absorbed dose to blood, calculated from external dose rate measurements using gamma probes positioned in the ceiling: patients with absorbed dose to blood exceeding 0.35 Gy (144 patients) had a significantly higher probability of successful ablation (63.9% ) than the other 305 patients (ablation rate 53.1%, *p* = 0.03), with a dose–response curve showing an increasing trend with the increase of the absorbed dose to blood. In contrast, no significant dependence of the ablation rate on the administered activity was observed. It is also a matter of fact that the high effectiveness of remnant ablation treatment using fixed activity, the absence of toxicity (apart from sporadic sialadenitis, still unpredictable with dosimetry), and the general technical and logistic difficulties for biokinetics studies and mass assessments of the target inhibited the spread of a treatment planning approach to thyroid remnant ablation, leading many to consider that it was not strictly necessary for treatment success. As regards locally advanced disease (neck nodes and metastases), hematological toxicity problems were scarcely observed using the currently fixed regimen of administration: administration of 7.4 GBq causes the safety threshold of 2 Gy to blood to be exceeded in 11% of cases, and this percentage can vary from 4% to 28%, considering the associated 95% confidence interval [145]. By contrast, hematological toxicity problems can be observed more frequently in patients with distant metastases: administration of 11 GBq could cause to exceed the safety threshold of 2 Gy to blood in 22% of patients, with the associated 95% confidence interval between 6% and 53. The outcome still spans from a complete response to death in the therapy of distant metastases: about 15% of patients with high-risk DTC have a significantly reduced life expectancy as many do not respond sufficiently to ^131^I therapy to prevent recurrence and progression of DTC, or even death [146]. In these cases, there would be important margins for improvement obtainable with dosimetry.

### 6.4. The Maximum Tolerable Activity

To increase therapeutic effectiveness in life-threatening diseases (e.g., pluri-metastatic DTC, liver malignancies, neuroendocrine tumors, neuroblastoma), overcoming the intrinsic limitations in spatial resolution of nuclear medicine imaging, a different approach may be to perform pre-treatment dosimetry to administrate the maximum tolerable activity (MTA) corresponding to the maximum tolerable absorbed dose to the OARs [114,133]. This approach of “Maximization” is based on the clinical observation that fewer treatments with higher absorbed doses delivered to lesions offer an increased chance of therapeutic success, because repeated treatments at lower absorbed doses could make metastatic tumors radioresistant [140,147]. MTA approach has long been adopted in NMT with pluri-metastatic and other advanced cases [33,78,114,138,140,148,149], as in these patients there is a strong need to improve the treatment effectiveness obtainable with fixed activities, and treatment planning is needed. Hematological toxicity is the side effect to be prevented in most DTC treatments, so the maximum tolerated activity is determined following the classical Benua-Leeper approach, i.e., in order not to exceed 2 Gy to the blood, as a surrogate of the absorbed dose to the bone marrow [150]. The EANM Dosimetry Committee formulated specific standard operational procedures to tailor the therapeutic activity to be administered for systemic treatment of DTC [81], such that the absorbed dose to the blood does not exceed 2 Gy and, at 48 h after administration, the whole-body retention does not exceed 4.4 or 3 GBq in the absence or presence of iodine avid diffuse lung metastases, respectively [80,150]: low activities of 131I NaI are administered pre-therapeutically followed by a series of blood and whole-body measurements. Implicit in all such approaches is the biokinetics assessed pre-treatment with a tracer activity and that of the in-treatment therapeutic administration are the same. To verify this assumption, a recent study [151] compared red marrow and blood absorbed dose values obtained pre-treatment with that obtained in-treatment for a group of 50 patients treated for metastatic thyroid cancer, using different dosimetric approaches: pre-treatment and in-treatment absorbed dose values to blood and red marrow appeared to be well correlated irrespective of the dosimetric approach used. In a retrospective analysis of 124 metastatic DTC patients who underwent dosimetric evaluation over a period of 15 y, the MTA approach was adopted using a risk-adapted approach [122]: the maximum administered activity was limited with radiation doses of 3 Gy to the red marrow or 30 Gy to the lungs, both limiting values corresponding to the absorbed dose of the external radiation therapy which is associated with a 5% risk of severe damage within 5 y. Eighty-three patients received 104 treatments, of which 13 were given for postsurgical ablation of thyroid remnants, 41 were given with curative intent, and 50 were for palliation (for the others, treatment resulted not indicated). The dose-limiting organ was red marrow in 19 of 41 treatments (46%) and lungs in 4 of 41 treatments (9.8%), whereas in the remaining 18 treatments (44%), the therapeutic endpoint of achieving a dose to the metastases of ≥100 Gy was reached delivering less than 3 Gy to red marrow or than 30 Gy to lungs. The administered activity corresponding to the red marrow absorbed dose of 3 Gy ranged from 7.4 to 37.9 GBq (mean, 22.1 GBq). The calculated doses to metastases ranged from 100 to more than 1000 Gy. Increased effectiveness for the MTA approach in terms of complete response as resulting by Response evaluation criteria in solid tumors (RECIST) was reported by Klubo-Gwiezdzinska et al. [81]. In a retrospective study examining 87 patients, 43 were treated with maximization approach and 44 with conventional fixed activities. The multivariate analysis, controlling for age, gender, and status of metastases revealed that the maximized activity group tended to be 70% less likely to progress (odds ratio, 0.29; 95% confidence interval, 0.087–1.02; *p* = 0.052) and more likely to obtain complete response (CR) compared to the fixed activity group (odds ratio, 8.2; 95% confidence interval, 1.2–53.5; *p* = 0.029). For the therapy of locally advanced disease, the MTA approach evidenced a significantly higher rate of CR in the maximized activity group compared with the fixed activity group (35.7% vs. 3.3%, *p* = 0.009. It was evident in the maximized activity group an indication of increased Progression Free Survival, but without achieving statistical significance. Conversely, similar findings were not obtained for the group of patients with distant metastases.

### 6.5. Recombinant Human(rh) TSH

Recombinant human(rh) TSH is worth noting also in the framework of absorbed dose driven approach in NMT: it has been developed for exogenous TSH stimulation in thyroid cancer patients remaining on thyroid hormone therapy. Clinical studies have shown that administration of rhTSH promotes radioiodine uptake and thyroglobulin production by thyroid cells with comparable efficacy to hypothyroidism for diagnosing residual or recurrent cancer [152], and it was used as a preparation for post-operative thyroid remnant ablation [153,154,155]. An international multi-center prospective, controlled, randomized, comparative study [156] involving 63 patients showed that the effective half-time in the remnant thyroid tissue was significantly longer after rhTSH than by thyroid hormone withdrawal (67.6 ± 48.8 h vs. 48.0 ± 52.6 h, respectively; *p* = 0.01) and that the specific absorbed dose to the blood was significantly lower (*p* < 0.0001) after administration of rhTSH (mean, 0.109 ± 0.028 mGy/MBq; maximum, 0.18 mGy/MBq) than after thyroid hormone withdrawal (mean, 0.167 ± 0.061 mGy/MBq; maximum, 0.35 mGy/MBq), indicating that higher activities of radioiodine might be safely administered after exogenous stimulation with rhTSH. Analogously, in a more recent study examining a group of 124 patients divided into subgroups with regard to modes of TSH stimulation (endogenous n = 100 versus exogenous n = 24), significantly higher blood doses were found in the subgroup with endogenous TSH stimulation (median 0.08 Gy/GBq) than in patients with exogenous TSH stimulation (0.06 Gy/GBq) [139].

### 6.6. ^124^I-PET

The possibilities introduced by the coming of ^124^I-PET deserve special mention. The work of Sgouros et al. was the first publication describing a rigorous approach to perform ^124^I-based treatment planning with Monte Carlo simulations, obtaining spatial distributions of absorbed dose and dose-volume histograms for a total of 56 tumors [157]. The mean absorbed dose values for tumors ranged from 1.2 to 540 Gy, the absorbed dose distributions presented wide variations ranging from a minimum of 0.3 Gy to a maximum of 4000 Gy, showing this high variability even within one patient. The use of conventional radiopharmaceuticals labeled with ^123^I or ^131^I for biokinetics studies presents several limitations, such as short half-time and limited availability for the former, worst quality of clinical images, and the risk of a “stunning” effect for the latter, whereas the recent availability of a PET tracer such as ^124^I has brought new perspectives for the dosimetric optimization. Due to the complex decay process of the ^124^I, more efforts for quantitative PET imaging are required with respect to 18F, to quantify the activity content in lesions for different background activities, with adequate spatial resolution and linearity [158]. The study by Capoccetti et al. aimed to evaluate the suitability of ^131^I-PET/CT for staging before radioiodine therapy, optimizing the activity administration for remnant ablation, and obtaining individualized dosimetry for patients with multiple distant metastases [159]. The MTA in two cases showed values of 13 and 9 GBq, whereas for 16 of 21 lesions studied the absorbed dose was lower than 80 Gy. The authors concluded that ^124^I-PET/CT could be routinely employed to obtain reliable dosimetry in patients with multiple metastases or to stimulate consideration of alternative therapies, thereby finally improving the management of those patients; however, its use is restricted by the limited availability. Additionally, in pediatrics, an interesting study highlighted how safe and informative this technique is: they found an absorbed dose to lesion with a range of 59–648 Gy/GBq; measuring blood samples with a well counter and whole-body clearance with an uncollimated NaI detector, they evaluated that the ^131^I MTA (aiming to avoid the >2 Gy blood absorbed dose) ranged from 19 to 42 GBq [160].

#### 6.6.1. ^124^I-PET for Salivary Glands

The first comprehensive approach to normal organ dosimetry using a quantitative tomographic imaging modality was performed with ^124^I-PET/CT by Kolbert et al. [161]. Calculations were done using a 3D voxel-based method: the highest mean absorbed dose (0.26 Gy/GBq) was obtained for the right submandibular gland, whereas the lowest mean absorbed dose (0.029 Gy/GBq) was referred to the brain. It is noteworthy that this publication showed good agreement between the absorbed dose to the heart chamber and the absorbed dose to blood obtained with the method of Benua et al. [80], so confirming the validity of the latter methodology. ^124^I-PET/CT allowed us to also focus better on the issue of salivary gland impairment following radioiodine therapy of DTC with high activity, a severe side effect highly affecting the quality of life. Jentzen et al. studied ten DTC patients, reporting an absorbed dose range for the submandibular and parotid glands of (0.18–0.55) Gy/GBq and (0.13–0.46) Gy/GBq, respectively, so confirming previous dosimetric data calculated by ^131^I planar gamma-camera scintigraphy and ultrasonography [162] and providing evidence that the average organ dose per administered activity is too low to state that the observed side effects to the salivary glands are induced by radiation [163]. The authors finally stated that a voxel-based calculation approach accounting for the non-uniform activity distributions in the glands is needed to clarify if the salivary gland damage could be considered radiation induced.

#### 6.6.2. ^124^I-PET for Thyroid Remnant and Metastases

The better image quality of ^124^I-PET images allowed more accurate dosimetric studies for thyroid remnants and small lesions. Freudenberg et al. [160,164] studied the remnant radioiodine kinetics through ^124^I PET/CT comparing the remnant absorbed dose in Gy/GBq under two stimulation methods: hypothyroidism (by thyroid hormone withdrawal) versus euthyroidism after the use of rhTSH. Published data suggested that the two stimulation methods do not produce significant differences in remnant absorbed dose. Analogous results were also obtained for metastases, using ^124^I-PET dosimetry [165]. However, further confirmative studies would be needed, due to the small sample size examined, and preferably multicentric clinical trials. ^124^I-PET/CT also encouraged studies on dose–response correlation for thyroid remnants and metastases. For instance, it is worth noting the study by Jentzen et al. [119], assessing the dose–response relationship in a large number of patients. The response rate was calculated based on lesions that received an average absorbed dose above the generally accepted threshold of 85 Gy for metastases and 300 Gy for thyroid remnants and was expressed as the percentage of completely responding lesions. Considering only target tissues that were amenable to reliable volume estimation, response rates were 63%, 88%, and 90% for lymph node metastases, pulmonary metastases, and thyroid remnants, respectively. For thyroid remnants, the response rate of 90% was in good agreement with the value obtained by Maxon (86%), whereas the response rate for lymph node metastases was only 63%, apparently lower than the corresponding Maxon’s value (81%) [118]. The authors explained that this discrepancy in the response rates of lymph node metastases was, in part, caused by the sensitivity difference between PET and planar scintigraphy systems. Second, the follow-up times in the study by Maxon et al. were approximately at least 10 months, sensibly higher than that adopted by Jentzen et al. [119], and this hypothesis may be further supported considering that DTC metastases are slow-growing tumors and that damaged cells can survive for months whereas radiation-induced cell death takes various routes, such as apoptosis and mitotic death. On the other hand, a more recent study by Wierts et al. using receiver-operating-characteristic curve analysis showed that for the known-volume group of targets, pretherapeutic ^124^I PET/CT lesion dosimetry can be used as a prognostic tool to predict lesion-based ^131^I therapy response with an area under the curve of 0.76 for remnants and 0.97 for metastases [166]. The corresponding lesion absorbed dose threshold value maximizing correct complete response prediction was 90 Gy for remnants and 40 Gy for metastases, so lower values than the accepted thresholds of 300 and 85 Gy. The authors explained that this difference may be due to difference in the analysis methodology (optimum thresholds defined as values that give the maximum correct classification combining both sensitivity and specificity) and that in DTC patient management, a correct prediction of incompletely responding lesions is important; consequently, for patient’s management decision making, higher absorbed dose threshold values may be preferred.

## 7. Conclusions

DTC represents more than 95% of cases of thyroid carcinomas and has excellent outcomes with prompt treatments. RAIT is a suitable choice for the irradiation of thyroid remnants, microscopic DTC, other nonresectable or incompletely resectable DTC, or all the cited purposes, and as adjuvant treatment in locoregional metastasis and distant metastasis. Dosimetry represents a valid tool that permits a tailored therapy to be obtained, sparing healthy tissue and so minimizing potential damages to risk organs. The advent of hybrid imaging such as SPECT/CT and PET/CT has given the possibility to better measure patient activity distribution thanks to an ameliorated anatomic definition. Absorbed dose represents a reliable indicator of biological response due to its correlation to tissue irradiation effects. It is a guide for treatment planning, mandatory for European regulations since it gives the knowledge of dose thresholds beyond which side effects shall be possible. The generally accepted thresholds for lesion-absorbed dose are 300 Gy for thyroid remnants and 85 Gy for metastases. Absorbed dose to the blood of 2 Gy is a limit for hematological toxicity and, as established by EANM Dosimetry Committee, the limitation of whole-body retention at 48 h after administration should be 4.4 Gy in the absence of iodine avid diffuse lung metastases and 3 Gy in their presence. Indeed, the absorbed dose should be below 30 Gy to avoid lung fibrosis. Small lesions are not so well detected, and the limited spatial resolution of emission tomography determines partial volume effects that also influences the estimation of cumulated activity and target dosimetry. The solution for this issue could be the use of ^124^I-PET images that allow for more accurate dosimetric studies for thyroid remnants and small lesions thanks to a higher spatial resolution.

## Figures and Tables

**Table 1 diagnostics-12-01763-t001:** Risk of cancer-related death for DTC by TMN classification.

Class	Age	T	N	M	Percentage Risk
I	<55 years	Any	Any	M0	<2%
II		Any	Any	M1	5%
II	≥55 years	T1/T2	N1	M0	5%
II	≥55 years	T3a/T3b	Any	M0	5%
III	≥55 years	T4a	Any	M0	5–20%
IVa	≥55 years	T4b	Any	M0	>50%
IVb		Any	Any	M1	>80%

**Table 2 diagnostics-12-01763-t002:** ATA risk of recurrence classification and corresponding ATA recommendation for RAIT from 2015 ATA Management Guidelines Task Force on Thyroid Nodules and Differentiated Thyroid Cancer.

Class	Risk of Recurrence	Histology	ATA Recommendation for Therapy
low	<5%	PTC and well-differentiated and minimally invasive FTC and No vascular invasion, no local and distant metastasis	Not recommended
intermediate	5–20%	Aggressive histology or vascular invasion (e.g., tall cell, insular, columnar cell carcinoma, Hurthle cell carcinoma, follicular thyroid cancer) Or Microscopic invasion into the perithyroidal soft tissues Or Cervical lymph node metastases Or ^131^I uptake outside the thyroid bed on the post-treatment scan	Recommended
high	>20%	Macroscopic tumor invasion Or Incomplete tumor resection with gross residual disease Or Distant metastasis	Recommended

**Table 3 diagnostics-12-01763-t003:** EANM Indications, contraindications, and relative contraindications for RAIT.

Indications	Contraindications	Relative Contraindications
adjuvant treatment post-surgery of persistent or recurrent iodine avid lesions (DTC)	Pregnancy and breastfeeding (absolute contraindication)	bone marrow depression
less differentiated tumor histotypes, such as papillary tall-cell, columnar cell or diffuse sclerosing or follicular widely invasive, poorly differentiated, or Hürthle cell	unifocal papillary thyroid cancer ≤1 cm sized without metastasis	pulmonary function restriction
patients > 45 years	thyroid capsule invasion	salivary gland function restriction
Intolerance to surgery	history of radiation exposure	neurological symptoms
Intolerance to other therapies (e.g., chemotherapy)	tall-cell, columnar cell, or diffuse sclerosing histotypes	Neurological damage (edema)

## Data Availability

Not applicable.

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
