# Peer review of "Personalized Dosimetry in the Context of Radioiodine Therapy for Differentiated Thyroid Cancer"

_diagnostics, 2022, doi:10.3390/diagnostics12071763_

Round 1
Reviewer 1 Report
Manuscript summarizes literature on the radioiodine therapy in differentiated thyroid cancer. Authors emphasize the need for implementation of personalized dosimetry approach that includes dosimetry-based calculation of the activity to be administered based on the patient specific biokinetics of the organs and lesions. Abstract and conclusions section were clearly written.
General comment:
It is not clear whether the authors provided version of the text that was not the final version, since there are highlighted parts of the text (with or without some corrections), or the text was highlighted after the submission.
Other comments:
Sections and subsections consist of one single paragraph which make reading difficult. Paragraphs are too long and need to be divided into several paragraphs to improve clarity of the text (Sections 4, 5, 6). Also, sentences with more than 40 or 50 words should be divided into several shorter sentences.
Line 34: Please check correctness of the sentence since there are other cancers that were more frequent in women than DTC in 2020 (breast, lung, colorectal, cervix uteri (GLOBOCAN 2020 source as a reference)).
Please add known risk factors for DTC in the introduction.
Editorial:
Please put mass number of the element in the superscript (for example 131I instead of the 131I, 124I, etc.)
line 63: change V classes to five classes
line 64: write what abbreviations stand for (T, N, M) first time when you introduce them in the text
line 142: ending of the sentence is not clear
line 223-238: also add activities in Bq as SI unit.
line 499: use of “In science and consciousness” is not clear.
line 628: write the full phrase with abbreviation RECIST in the brackets.
line 627-634: split the sentence into several shorter sentences.
Author Response
I want to thank the referees for the attention shown and the precious suggestions made, which we've welcomed. We have been glad to put them in place, conscious of the fact that your advices and suggestions will certainly improve the quality and comprehensibility of our work.
All recommended corrections have been made and underlined in the manuscript, as detailed below:
All corrections made to the text have been highlighted and underlined in yellow.
Regarding the reviewer's comment:
1) We have added the following paragraphs:
4.1 Bone marrow deficiency
4.3 Sexual sphere side effects
4.4 Second recurrences
4.5 The stunning fenomenon
4.6 Reduced kidney function and dosimetry considerations
6.1 Bone marrow and lung issues
6.6.1 124I-PET for salivary glands
6.6.2 124I-PET for thyroid remnant and metastases
2) We reported the GLOBOCAN 2020 citation
3) We add risk factors for DTC (line 50-69: “Even if the causes of thyroid cancer are unknown, many studies evaluated risk factors…Estradiol is seen as a stimulator for benign and malignant neoplasm”)
Editorial:
1) We put mass number of the element in the superscript;
2) line 91: change V classes to five classes
3) line 107: we have written what abbreviations stand for TNM
4) line 172: we clarified the phrase
5) line 260-301: we add activities in Bq as SI unit.
6) line 539: we removed “In science and consciousness”.
7) line 669: we have written the full phrase with abbreviation RECIST in the brackets.
8) line 672-680: we have split the sentence into shorter sentences.
An English review of the entire manuscript was carried out by a native speaker and appropriate corrections were made
Reviewer 2 Report
The manuscript is very interesting in its objective, but it needs some improvement in definitions (as Hurtle tumors, some do not characterize them as lined 33-37 and 42) and the tumor stage should be stuck to the last ATA and European guidelines. Moreover, the most interesting part: dosimetry should be separated per tumor. I understand most documents are related cases but then I suggest compiling them so the reading turns interesting and understandable. A good English revision is necessary.
Author Response
I want to thank the referees for the attention shown and the precious suggestions made, which we've welcomed. We have been glad to put them in place, conscious of the fact that your advices and suggestions will certainly improve the quality and comprehensibility of our work.
All recommended corrections have been made and underlined in the manuscript, as detailed below:
All corrections made to the text have been highlighted and underlined in yellow.
Regarding the reviewer's comment:
1) We add the definition of Hurtle tumor;
2) An English review of the entire manuscript was carried out by a native speaker and appropriate corrections were made.
3) The tumor stage already complies with the latest ATA 2015 and European 2008 guidelines as reported in the text
Round 2
Reviewer 1 Report
Line 163-173
Definition of the GBq in the brackets is not correct. 1 GBq is 109 Bq and not 1010 Bq as written. It would be best to delete interval of the activities in the brackets because it is clear what GBq means. If not, correct the values in the brackets.
Author Response
Thank you for this clarification and I have modified as suggested

This manuscript is a resubmission of an earlier submission. The following is a list of the peer review reports and author responses from that submission.
Round 1
Reviewer 1 Report
The paper is a review paper and I believe as a review paper the manuscript doesn't add any sort of significant output for the research community. The manuscript completely lacks the comparative analysis between existing techniques to tackle the research problem. I believe the authors have just represented one side of the problem and demonstrating it to be a bigger issue while no discussion is made on existing techniques. Further, the paper is just representing the literature leaving many aspects behind. The authors need to understand the difference between literature review and review manuscript.
Reviewer 2 Report
This paper is the result of a lot of pulling together of evidence about the complexities of optimal treatment in DTC, the disease itself , and dosimetric approaches to treatment
However in its present form it is almost unreadable and does not demonstrate the authors knowledge or intensions clearly.
Many of the sections describe previous classifications and outcomes eg (the section on risk classification) that would make far better reading as carefully composed tables.
The entire text is unformatted and has no paragraphs to break up the details, which can jump about A thorough thinking through and rewriting with careful use of subsections and organisation is needed to make this detail come alive. This work needs to be lucid and clear.
The section on dosimetry does not demonstrate the authors have an understanding of the very latest thoughts and techniques, although there is an attempt to argue some of the complexities involved. I would direct them to the MEDIRAD project.
Although any attempt to publish in a second language is to be praised, this paper struggles with archaic or uncommon, or plainly ungrammatical English, and will need rewriting under the guidance of a native English speaker. This would help give the detailed work its best chance to be shown.